# Access to Land for Agricultural Entrepreneurial Activities in the Context of Sustainable Food Production in Borgou, according to Land Law in Benin

**Bienvenu Dagoudo Akowedaho** [1,2,*] **, Inoussa Guinin Asso** [3,4]**, Bruno Charles Pierre O'heix** [4]**, Soulé Akinhola Adéchian** [3] **and Mohamed Nasser Baco** [3]

[1] Faculty of Agriculture, Uganda Martyrs University, Nkozi and Uganda, Kampala P.O. Box 5498, Uganda
[2] Department of Agricultural Economics and Rural Sociology, Faculty of Agronomy, University of Parakou, Parakou P.O. Box 123, Benin
[3] Laboratoire Société–Environnement (LaSEn), Department of Agricultural Economics and Rural Sociology, Faculty of Agronomy, University of Parakou, Parakou P.O. Box 123, Benin
[4] GIZ, Projet Promotion d'une Politique Foncière Responsable (ProPFR), Parakou P.O. Box 123, Benin
[*] Correspondence: bienvenu.akowedaho@stud.umu.ac.ug; Tel.: +229-96659847

**Abstract:** Access to land is crucial for food systems to address the challenges caused by habitat and biodiversity loss, land and water degradation, and greenhouse gas emissions. Sustainable food production requires land security upstream for agricultural production. Land security emanates from the land law implemented in-country by government policy. In the span of a decade (2007–2017), three different land reforms have been adopted in Benin. This paper aims to investigate the land rights and land tenure security for sustainable food production according to land law and the factors that influence agricultural entrepreneurial activities in North Benin. The study was carried out in the Borgou department, mainly in five communes that are beneficiaries of the Responsible Land Policy Project of GIZ (Promotion d'une Politique Foncière Responsable: ProPFR/GIZ). A multistage sampling procedure was used to select the agricultural entrepreneur respondents. A total of 102 agriculture entrepreneurs were interviewed in 25 villages. According to land law in Benin, the results highlight the different levels of land tenure security and land rights represented by types of land documents: type contract (use right), certificates of customary ownership (ADC), and land title. The research reveals that 44.3% of the land of agriculture entrepreneurs' respondents possessed the certificates of customary ownership and 18% possessed the land title. The facilitation of access to legal land documents such as certificates of customary ownership and land titles can protect agricultural entrepreneurship for sustainable food production.

**Keywords:** land access; agricultural entrepreneurship; land law; sustainable production; Benin

## 1. Introduction

In Africa, agriculture and food systems are an important provider of employment and livelihoods [1]. The global food system may be facing serious problems, but many powerful actors are hard at work to address the challenge without too disruptive a change [2]. Food systems across the world have caused habitat and biodiversity loss, land and water degradation, and greenhouse gas emissions [3]. Land use for food production is a key component of land systems and a central part of food systems at the interface between actors that provide inputs in food systems and those that process and consume food [4]. The intersections between land and food systems are key to many global environmental changes, environmental justice, and sustainability challenges [4]. Access to land is crucial to the improvement of agricultural performance, food security, and economic development [5,6]. Land access is defined by the land tenure system of each country. The land tenure system can be defined as the rights and institutions that govern access to and use of land [7]. The

tenure system of land involves a system of rights, duties, and responsibilities concerning the use, transfer, alienation, and ownership security of land and its resources [8].

The Republic of Benin is a former colony of France which gained independence in 1960. Benin became, in 1991, a democratic republic. Like some countries in Sub-Saharan Africa, the land system of Benin has undergone modification of proprietary rights. Since 1991, the land system has been characterized by diverse actors, including traditional community leaders, families, lawyers, and the government, which implemented dual laws (customary law and modern law). This situation creates insecurity in agricultural entrepreneurial activities, which negatively influences agriculture investment for food production. The land policy reforms were initiated by different governments and legislation. The last land policy reform was the legal foundation of the land administration in Benin. This is the 2013 Land Administration Law [9] that, together with the 2017 addendum [10], replaced different previous land laws (Table 1). The new land law encompasses, from art. 347 to art. 378, the customary and rural land rights and the justice and administrative functions of rural land management [9].

**Table 1.** Chronological evolution of land policies in Benin Republic after 1990.

| Important Dates | Type of Legislation | Description of Consecutive Legislations |
|---|---|---|
| 1990 | Constitution | N° 65-25: Review of the land ownership of the Constitutional Marxist military regime established in 1972 |
| February 2007 | Post-colonial land | Rural land law established |
| 14 August 2013 | Laws | Land law N° 2013-01 established |
| 15 August 2017 | New laws modifying land law N° 2013-01 | Land law N° 2017-15 established |

The execution of the land administration in Benin is assigned to the National Land Registry and Agency (*Agence Nationale du Domaine et du Foncier*: *ANDF)*. The Land Administration Law and the *ANDF* have the objective of centralizing land administration and recording the entire national territory in one digital central land administration system. Generally, land tenure reform has made a great contribution to improving agricultural productivity and can provide an effective long-term solution to food security [11]. The condition of access to land is crucial for agricultural entrepreneurial activities. The demands, opportunities, and challenges of the changing business environment in the agricultural industry have required farmers to become entrepreneurial [12]. However, investment for food production through agricultural entrepreneurial activities depends on the land tenure security. The authors of [13] found that, although the links between tenure security and agricultural productivity are of primary interest, tenure security is endogenous, and a positive correlation between investment and land tenure security could occur because people invest to become more tenure secure. According to [14], individual and secure land tenure rights are vital components of a productive agricultural sector, which is crucial to poverty alleviation and economic growth. Entrepreneurship is one of the determinants of economic growth [15], and the creation of small and medium-sized enterprises, especially in the secondary and tertiary sectors, can resolve problems induced by the changes in agriculture [16]. In developing countries such as Benin, land issues are of crucial importance due to the significant incidence of agriculture on economic growth [17], and agricultural entrepreneurial growth impacts poverty significantly [18]. The importance of entrepreneurship in agriculture means that interest in this field of research has only gained more interest recently and is still being consolidated [19]. This article aims to analyze the land rights and the land tenure security for agricultural and entrepreneurial activity according to the new land law in Benin. Furthermore, the types of land documents provide by the land law in Benin are examined according to the land rights and the level of land tenure security [20] in one part, and according to the factors influencing agricultural entrepreneurial activities in the other.

## 2. Methods and Materials

### 2.1. Method for Collecting and Analyzing Data

2.1.1. Study Area

The study was carried out in the Borgou department (Figure 1). Located in northern Benin, the territory covering the department of Borgou is 25,856 km$^2$ with 1,214,249 inhabitants [21], which represents 46% of Benin's surface area, and is situated between latitude 8°45′ and 12°30′ N and between longitude 2° and 3°15′ E. This region is characterized by a Sudano-Guinean climate with a rainy, relatively cold season from April to October and a dry, very hot season from November to March. Annual rainfall varies from 900 to 1200 mm, and extreme average temperatures are 30.8 °C in February (dry season) and 24.4 °C in August (rainy season). The department has a total of eight communes, of which five communes represent this study area: Bembèrèkè, Kalalé, N'Dali, Sinendé, and Tchaourou.

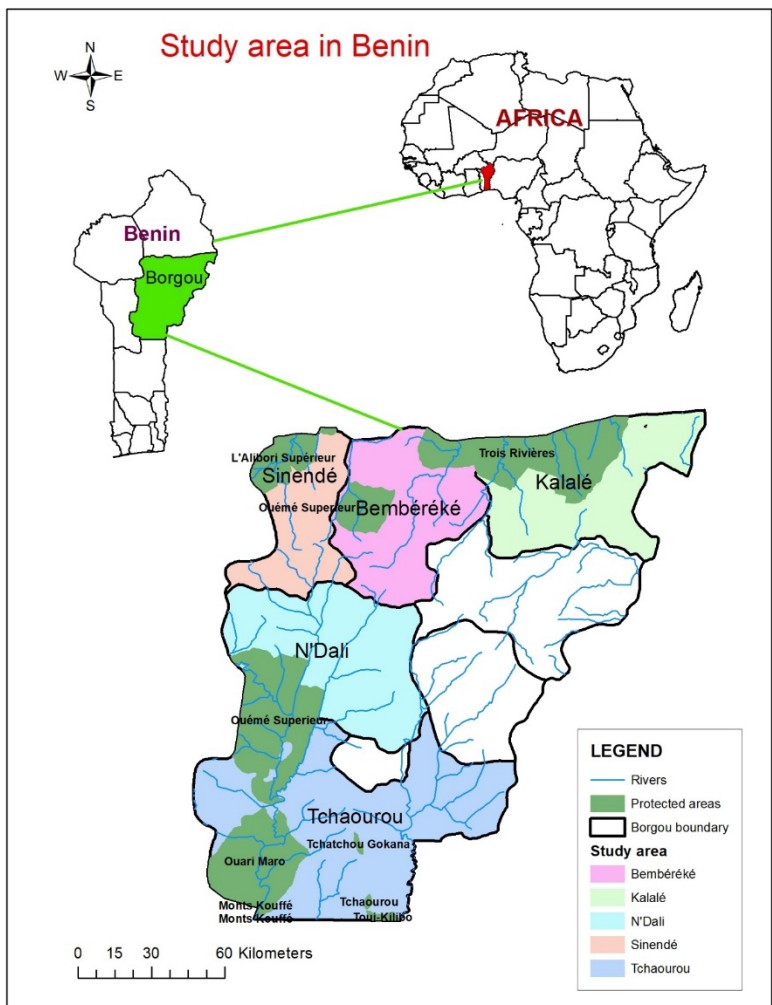

**Figure 1.** Study area.

2.1.2. Data and Sampling Procedure

The data used in this study come from the secondary data, i.e., the Land and Domain Code law (Code Foncier et Domanial) reviews in 2017, and the primary data collected through the empirical survey. Land tenure security refers to the actual control of land [22]. It mainly depends on "the extent to which legal rules are effectively enforced at the local level" [23]. The legal rules are defined in the Land and Domain Code, which is the secondary database to analyze the different levels of land security for agricultural entrepreneurial activities through the possessing of land papers.

The empirical survey was focused on the types of land documents possessed by the agriculture entrepreneurs and their land tenure security following the domain and code land law in Benin. The survey also examines the outputs of agricultural entrepreneurial activities (activity sector, land harness, job creation, etc.), as well as the factors influencing this activity. It was carried out in November 2020 with the collaboration of the GIZ institution.

A multistage sampling procedure was used to select the agricultural entrepreneur's interviews. In the first stage, the communes of N'Dali, Bembèrèkè, Sinendé, Tchaourou, and Kalalé were purposively selected as the five communes that are beneficiaries of the Responsible Land Policy Project of GIZ (Promotion d'une Politique Foncière Responsible: ProPFR/GIZ). This project aims to improve the institutional framework and processes for securing land use and ownership rights in the department of Borgou [24]. Second, we asked the GIZ officer to provide us with a list of villages in their locations. From the lists, we randomly selected five villages per commune. Therefore, a total of 25 villages were selected. Third, we acquired the lists of the agricultural entrepreneurs in all the selected villages and applied a proportional random sampling (average 20 agricultural entrepreneurs per commune) procedure to select individual farmers to be interviewed. A total of 102 agricultural entrepreneurs were interviewed in the five communes (Table 2).

**Table 2.** Study area and sampling.

| Communes | Villages | Sample Size |
|---|---|---|
| N'Dali | Tamarou, N'Dali centre, Sirarou, Mareborou, Boko | 21 |
| Bembèrèkè | Gahmaré, Bembèrèkè, Gamia, Wanrarou, Ina | 20 |
| Sinendé | Siki-Fô-bouré, Guessou-bani, Niaro-Gouou, FômBouko, Sékéré-Soka | 20 |
| Tchaourou | Tchaourou centre, Guinerou, Chala-Boyadi, Akoudanmon, Tekparou | 20 |
| Kalalé | Kidaroukperou, Kalalé-Nassiconzi, Nassimina, Peonga, Suanin-Dunkassa | 21 |
| **Total** | | **102** |

### 2.1.3. Data Collection and Analysis

In order to achieve the objective of this study, descriptive statistics, such as frequency counts, simple percentages, means, and tables, were used to analyze the results. They were all exported into the International Business Machines (IBM) Statistical Package for Social Scientists (SPSS) version 20 data editor for a one-sample *t*-test.

However, agricultural entrepreneurial activity is a complex issue, as the farmer can be an owner, a tenant, a manager, a subcontractor, or a combination, suggesting that the methods used to analyze business entrepreneurs in other sectors may not be easily transferred to an analysis of agricultural entrepreneurial activity. Therefore, the *t*-test revealed the significance of the factors through the null hypothesis (H0) and alternative hypothesis (H1). The factors that can influence the agricultural entrepreneurial activity identified in this study are land access mode factors (occupation, heritage, donation, and sharecropping) and socioeconomic factors (financial support from parents or friends, land availability, experience in the agriculture sector, microfinance or bank loans, autonomy decisions, and profitability of activity).

### 2.2. *Agriculture Entrepreneurial Activity*

Agriculture is basically the set of practices through which people produce food [25]. Agriculture is multifunctional and evolving [26,27]. Agricultural production cannot be separated from the other aspects of food systems, such as food supply chains, the food environment, and consumption [28,29]. The function of agriculture is not only the produc-

tion of food; it has also shaped landscapes, preserved biodiversity, and created a cultural heritage over centuries [30]. However, defining farmers' entrepreneurial activity is complex as they do not operate in similar business activities characterized by their urban counterparts [31], because entrepreneurship in the agricultural sector encompasses all aspects of food systems (food supply chains), biodiversity, environment, consumption, and tourism. Entrepreneurship is the process through which opportunities to create future goods and services are discovered, evaluated, and exploited [32]. Entrepreneurial options that farmers have employed include implementing selective product specialization, enterprise diversification, market orientation, production upscaling, product development, process innovation, and vertical integration [33]. Agriculture production and related activities (food processing, biodiversity, environmental protection, consumption, and tourism) serve as the foundation for agricultural entrepreneurial activities with the goal of creating added value. This encompasses the opportunities to create new goods and services or new value chains. For De Wolf, smallholder farming entrepreneurship includes the production of specialty food products for niche markets, the provision of services to other farmers, and the use of agricultural assets such as the farmhouse and the farm animals to attract paying visitors [34]. According to farm management studies, entrepreneurship by farmers promotes farm diversification [31,35]. Although farmers tend to establish a number of similar farm businesses, these businesses can still be seen as entrepreneurial because they require "contracts with new customers and/or suppliers, new marketing channels, and reorganization of the management in the business" [36] (p. 242). In this paper, agricultural entrepreneurial activities are defined as all activities based on the use of land to produce food for the market and preserve biodiversity.

*2.3. Conceptual Framework*

This article aims to position the concept of land tenure security in sustainable food production in Benin through agricultural entrepreneurial activities. Land investment in Africa is driven by the large amount of perceived available land and weak land rights [37], increased demand and prices for food, energy systems transitions, biodiversity conservation, climate change responses, geopolitics, and development strategies [38]. The land conflict affects land investment, which weakens agricultural entrepreneurial activities and sustainable food production. Land conflict is a result of deficiencies in arbitration mechanisms in contexts of plurality of norms [39] and competition between authorities [40], not just of increasing pressure on land. It also highlights the interest that powerful actors have in managing confusion [41], in urban and peri-urban environments [42], as well as in rural areas. Rural land is coming under multiple pressures which include population growth and increasing fragmentation, land-use conversion, commercial investments, environmental degradation due to drought, soil erosion, and nutrient depletion, as well as natural disasters and conflicts [20]. The degree of implementation of the reforms is even more crucial.

Benin's case is particularly interesting in this regard because the land debate began in the 1990s, and Benin implemented three different land reforms in the span of a decade (2007–2017). The last land reform (2017) aims to centralize land administration, with the objective of recording the entire national territory in one centralized digital cadaster (*"le cadastre national numérique"*). Land is an essential resource for agriculture and entrepreneurial activities. Access to land is a fundamental basis for human shelter, food production, and other economic activity, including by businesses and natural resource users of all kinds [20]. Therefore, the modes of access to the agricultural land in Benin (lease, purchase, testamentary succession, and inter vivo donation) were inscribed in land and domain code law (art. 360, 2013). Reviewed in 2017, the domain and code of land law specified that each village and each municipality must have a land administration committee (Land and Domain Code: art.428, 2017). However, the land tenure security for agricultural entrepreneurial activities involves some stakeholders, which facilitate the process of accessing the agricultural land. According to the land law in Benin, the stakeholders identified are land policy, institutions of law, local authorities, informal customary,

and partners such as nongovernmental organizations (NGOs), professional organizations (POs), and private sectors. Sustainable food production starts with land tenure security (an essential resource) and food market availability, which contribute to food security, job creation, and cash flow (Figure 2).

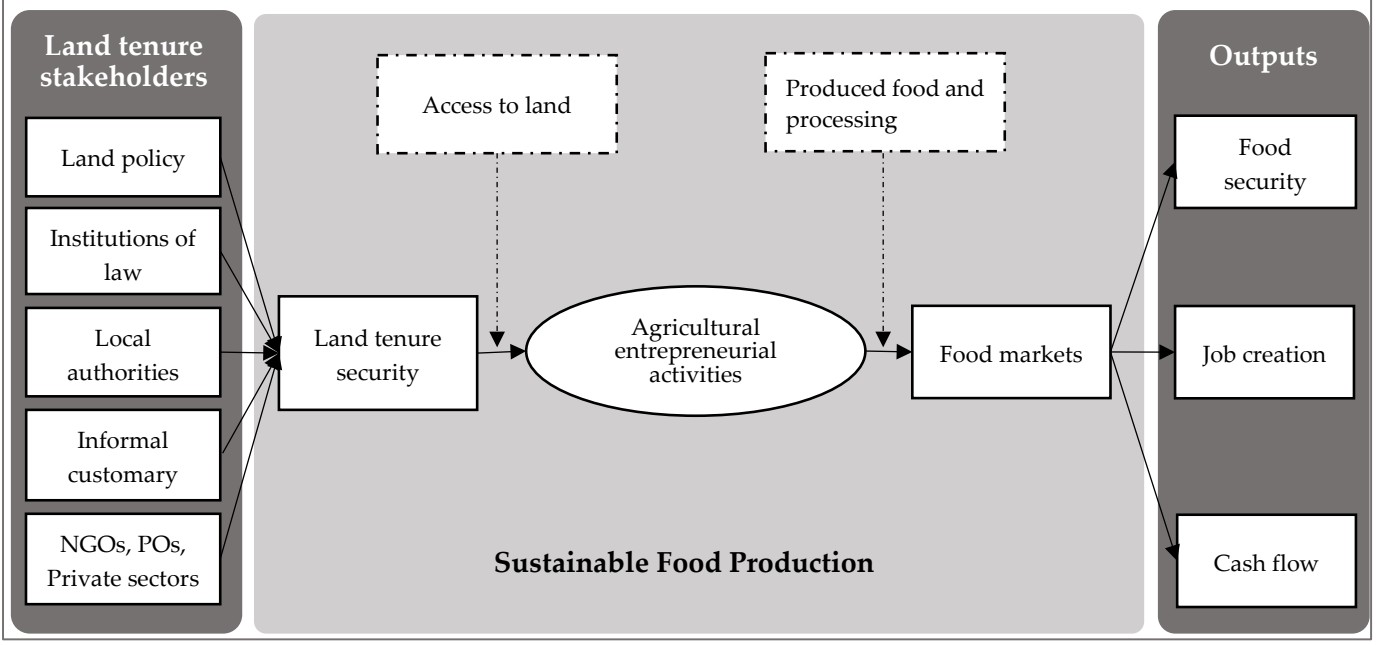

**Figure 2.** Conceptual framework.

## 3. Results and Discussion

*3.1. Land Rights and Land Tenure Security for Agricultural Entrepreneurship, according to Domain and Code Land Law in Benin*

Accessing land for agriculture activities is defined by the recent land reform that was enacted by the 2013 Code Foncier et Domanial (Domain and Code of Land law) reviews in 2017. According to the domain and code of land law (art.360) in Benin country, "*the permanent transfer of rural customary land may be by purchase, intestate or testamentary succession, inter vivo donation, or by any other effect of obligation . . . This contract must be based on the rural land title corresponding to the plot of land concerned, provided that the village in which it is located and which has been the subject of the establishment of a rural land tenure plan as provided for by this code*". Land is a veritable ingredient in agricultural entrepreneurial activities. The formalization of land rights for exerting agriculture activities in order to secure production necessitates a process which encompasses the landowners, as well as local and national authorities. The informal institution (the landowner's family) and the formal institution (local and national governance institutions) are both involved in this process. Actual tenure security, which could be derived from both formal and informal institutions, is based on tenure holders' "actual control of property, regardless of the legal status in which it is held" [22]. The informal institution is governed by customary law, which is recognized in the domain and code of land law of Benin (art. 351). This does not guarantee the security of land but constitutes the first step toward formalizing the land rights. However, agricultural activities may be carried out on land subject to customary regulation. When the customary regulation is in writing, the name of the contract type is taken (art. 354). This contract (mortgage, rents, pledges . . . ) must be registered in the village section of land tenure, with a copy to the commune's land management commission and the local office responsible for confirming land rights. The contract type for at least 10 years can be disrupted at any moment with several motifs, e.g., if the tenant violated the contractual clauses or repossession of the land by the owner with a view to exploitation.

The formalization land can provide the certificate of customary ownership (Attestation de Détention Coutumière: ADC). The obtention of this certificate is defined in the domain and code of land law in Benin (art. 352) by applying for ADC to the mayor of the commune. The mayor with the support of village section of land management (Section Village de gestion foncière) then proceeds to the contradictory public survey.

The record of the survey is transmitted to the mayor who, with favorable opinion of the neighborhood, establishes five copies of the ADC distributed as follows: one for the commune commission land management, one for the village section of land management, one for the land domain office, and one for the requestor.

This certificate of customary ownership can be contested by following the same process. According to the domain and code of land law, art. 386 to art. 393, the land conflict can be managed in a friendly manner or in front of the mayor. The land title confirms the ownership of land according to art. 376 of the law of the land and domain code in Benin. It is the uncontested right of the landowner (high-level land security).

According to the domain and code of land law, the land rights can be classified from informal to formal. The typology of land rights [20] along the horizontal axis is correlated with the types of land document in Benin. This typology of land rights (the land rights continuum) indicates the land value attributed by land law. The formality of land rights according to the type of land document shows the different level of land tenure security in the vertical axis of the proposed model.

One of the main uses of the continuum of land rights model is to understand relationships between land rights and land tenure security, with a focus on improving land tenure security, especially for the poor [43]. The land document types reflect the different levels of land tenure security. This land security is proposed on the vertical axis through the variables of legitimacy (acknowledgement by people) and legality (legislation is linked to policy).

Figure 3 depicts the land rights and the land tenure security for agricultural activities according to the domain and code of land law in Benin.

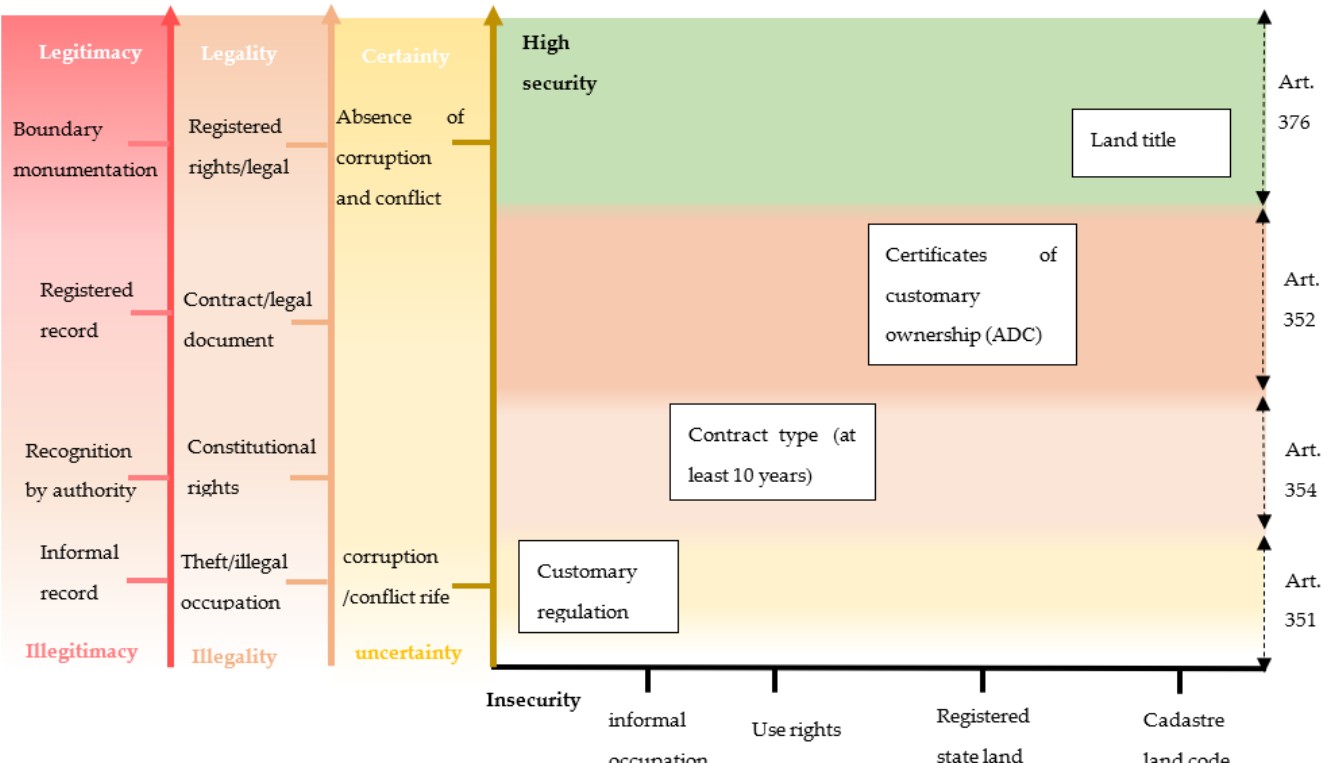

**Figure 3.** Land rights and land tenure security according to the domain and code of land law.

This diagram shows the different levels of land tenure security and the land rights for agriculture activities according to the types of land document possessed.

According to UN-Habitat (2018), tenure security still tends to be strictly defined in more statutory forms of legal documentation, such as individual land titles. However, the customary regulations recognized by the domain and code of land law (art. 354) do not guarantee the security of land for agricultural entrepreneurial activities. Since colonial times, customary land tenure was not thought to provide adequate tenure security, thereby discouraging investment and negatively affecting agricultural productivity [44,45]. The customary land tenure corresponds to the illegal occupation where there are the most conflicts and corruption. The contract type (art. 354) is recognized by authorities (legitimacy axe) and grants use rights to agriculture entrepreneurs. The certificate of customary ownership (art. 352) reflects the registered state land for the land rights. This land document provides agriculture entrepreneurship through the legal title and registered record for the land tenure security. The land title (art. 376) indicates the absence of corruption and conflicts. It is a high level of land tenure security that corresponds to agricultural entrepreneurship and security investment for sustainable food production.

### 3.2. Different Types of Land Documents Possessed by Agriculture Entrepreneurs

In Benin, land reform defined the process of formalizing agricultural land in order to achieve total land tenure security. This process, which involves the local, communal, and governmental authorities, requires financial resources from agriculture entrepreneurs. Certificates of customary ownership (ADC) range between 30,000 and 50,000 FCFA (47.55 and 79.25 USD) (1 USD = 630.853 FCFA on 15 May 2022), depending on the size of the plot. According to some authors, this is above the financial capacity of the farmers and will make it difficult for the rural people to access ADC [46]. The process of getting the land title at the National Land Registry and Agency depends on the communes and the size of the plot. According to [46], the cost for citizens is doubly limited. First, surveyors freely fix the cost of demarcation, which is one of the major costs of the procedure, and frequently exceeds 300,000 FCFA (475.55 USD). Second, the registration procedure follows several others (purchase, having a certificate, etc.), and the full cost for users must include those steps, as well as other indirect expenses (travel to the office, etc.). The cost of land titles remains high; however, above all, demarcation fees are unregulated and, therefore, freely fixed by surveyors. The introduction of notaries as a mandatory step for transfers incurs significant upstream costs: a notarial sales contract or a minute registration costs around 300,000 FCFA (475.55 USD). Following the different costs for land paper in the Borgou department, the percentage of land paper per commune is depicted in Figure 4.

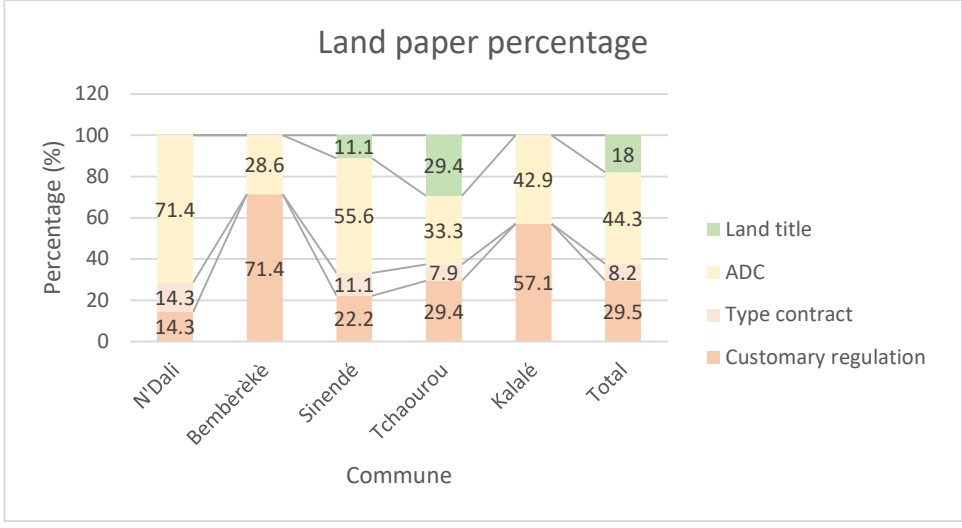

**Figure 4.** Types of land documents.

According to the domain and code of land law, 62.3% of the land of agriculture entrepreneurs' respondents was secure (44.3% possessed the certificates of customary ownership (ADC) and 18% possessed the land title). Land registration for ADC requires a survey with the landowner, the neighbors, local authorities (customary authorities, chief of village), and topographical markers, which makes the processing time long. However, this process of getting an ADC is organized by village or by zone, which facilitates the availability of all actors at the moment, according to the respondents. Amongst the five communes of the study, the number of agriculture entrepreneurs who possessed the land title in the communes of Tchaourou and Sinendé was 29.4% and 11.1%, respectively. More than 50% of agriculture entrepreneurs in the commune of Bembèrèkè and Kalalé still follow customary regulations such as land documents. More than 90% of the rural population in Sub-Saharan Africa (of which 370 million people are considered to be poor) accesses land a via legally insecure customary and informal tenure systems [47]. This shows the land insecurity for the agriculture entrepreneurs in both communes.

### 3.3. The Sociodemographics of the Agricultural Entrepreneurs

The findings from descriptive statistics provide insight into the sociodemographic characteristics of agricultural entrepreneurs' respondents. These sociodemographic characteristics of agricultural entrepreneurs group sex, education background, and experience. Table 3 provides the frequency and percentage of agricultural entrepreneur respondents.

**Table 3.** Agricultural entrepreneur profiles.

| Characteristics | | Communes | | | | | Total |
|---|---|---|---|---|---|---|---|
| | | Bembèrèkè | Kalalé | N'Dali | Sinendé | Tchaourou | |
| | | *N%* | *N%* | *N%* | *N%* | *N%* | *N%* |
| Sex | Female | 10.0 | 19.0 | 23.8 | 4.8 | 5.3 | 11.8 |
| | Male | 90.0 | 81.0 | 76.2 | 95.2 | 94.7 | 88.2 |
| Education background | Illiterate | 30.0 | 9.5 | 52.4 | 9.5 | 0.0 | 20.6 |
| | Primary | 15.0 | 23.8 | 4.8 | 42.9 | 52.6 | 27.5 |
| | Secondary | 30.0 | 47.6 | 28.6 | 47.6 | 42.1 | 39.2 |
| | University | 25.0 | 19.0 | 14.3 | 0.0 | 5.3 | 12.7 |
| Ethnic groups | Bariba | 95.2 | 75.0 | 95.2 | 0.0 | 19.0 | 57.8 |
| | Peulh | 0.0 | 15.0 | 0.0 | 0.0 | 23.8 | 7.8 |
| | Lokpa | 0.0 | 0.0 | 0.0 | 5.3 | 0.0 | 1.0 |
| | Nago | 0.0 | 0.0 | 4.8 | 89.5 | 57.1 | 29.2 |
| | Others | 4.8 | 10.0 | 0.0 | 5.3 | 0.0 | 3.9 |
| | | Mean | Mean | Mean | Mean | Mean | Mean |
| Age | | 48.05 (±11.82) | 41.48 (±5.68) | 48.19 (±10.05) | 42.62 (±10.46) | 46.89 (±9.26) | 45.39 (±9.90) |
| Experience | | 14.80 (±10.68) | 15.24 (±4.52) | 9.33 (±4.26) | 10.71 (±6.06) | 13.5 (±5.42) | 12.70 (±6.86) |

The profile of each respondent is a summary of general demographics for the entire sample. The majority of agricultural entrepreneurs (88.2%) were men. Almost 20.6% of agricultural entrepreneur respondents did not have an educational background (illiterate), 27.5% had completed the primary level, in contrast to secondary school (39.2%). The average age of agricultural entrepreneurs was 45.39 (±9.90) years. The mean agricultural entrepreneur experience was 12.70 (±6.86) years.

### 3.4. Different Sectors of Agricultural Entrepreneurial Activities

Agricultural entrepreneurial activities embrace all sectors of agriculture production and non-agriculture production. This research focuses on agricultural entrepreneurs who

have land issues and are registered in the GIZ database. Table 4 shows the percentage of entrepreneurs by agricultural entrepreneurial activity sector.

**Table 4.** Sectors of agricultural entrepreneurial activities.

| Sectors | | Communes | | | | | Total |
|---|---|---|---|---|---|---|---|
| | | N'Dali | Bembèrèkè | Sinendé | Tchaourou | Kalalé | |
| | | *N%* | *N%* | *N%* | *N%* | *N%* | *N%* |
| Vegetal production (annual crops) | Female | 23.8 | 0.0 | 4.8 | 5.3 | 30.0 | 11.8 |
| | Male | 76.2 | 100.0 | 95.2 | 94.7 | 70.0 | 88.2 |
| Vegetal production (perennial crops) | Female | 20.0 | 0.0 | 0.0 | 6.2 | 11.8 | 11.8 |
| | Male | 80.0 | 100.0 | 100.0 | 93.8 | 88.2 | 88.2 |
| Livestock | Female | 25.0 | 0.0 | 0.0 | 5.9 | 7.7 | 11.8 |
| | Male | 75.0 | 100.0 | 100.0 | 94.1 | 92.3 | 88.2 |
| Crop processing | Female | 0.0 | 25.0 | 100.0 | 0.0 | 60.0 | 11.8 |
| | Male | 100.0 | 75.0 | 0.0 | 0.0 | 40.0 | 88.2 |

Four sectors were identified as agricultural entrepreneurial activities in the area under study: vegetal production (annual crops), vegetal production (perennial crops), livestock, and agri-food transformation. In the Borgou department, the agricultural entrepreneurial activities are conducted by women, as well as men. The results show that, although 11.8% of agricultural entrepreneurs were women, in the communes of Bémbèrèkè and Sinendé, agricultural entrepreneurial activities such as vegetal production (perennial crops) and livestock were still completely performed by men. In the communes of N'Dali and Kalalé, women featured in entrepreneurship linked to vegetal production (annual crop), with 23.8% and 30.0%, respectively. Crop processing was an agricultural entrepreneurial activity performed only by women (100%) in the commune of Sinendé, as well as by the majority in the commune of Kalalé (60.0%). Women had more opportunities for crop processing in both communes than for other agricultural entrepreneurship activities. Benin's customary law, however, does not allow succession to land for women, and this limits women's ability to invest in agricultural entrepreneurship with insecure land. In Tchaourou, no agricultural entrepreneurs are involved in crop processing.

*3.5. Agricultural Entrepreneurs' Average Land Use and Employment*

In the context of agricultural entrepreneurial activities, the acreage of land used and the number of jobs created are important indicators. These indicators give an overview of the size and scope of the agricultural enterprise. Table 5 highlights the average land area and the average number of employees with their standard deviation.

**Table 5.** Average land area and number of employees.

| Indicators (Average) | Commune | | | | | Total |
|---|---|---|---|---|---|---|
| | N'Dali | Bembèrèkè | Sinendé | Tchaourou | Kalalé | |
| Land available (ha) | 15.55 (±13.90) | 16.42 (±15.00) | 29.48 (±16.53) | 14.26 (±14.36) | 12.57 (±9.85) | 17.74 (±15.117) |
| Land harnessed (ha) | 13.02 (±13.32) | 16.42 (±15.07) | 20.33 (±13.94) | 9.84 (±9.73) | 12.33 (±10.00) | 14.41 (±12.89) |
| Land secured (ha) | 11.67 (±10.97) | 12.35 (±18.02) | 8.83 (±6.77) | 11.73 (±10.54) | 10.24 (±12.68) | 11.16 (±12.296) |
| Permanent employees (male) | 4.48 (±4.29) | 2.75 (±5.69) | 4.76 (±3.66) | 3.84 (±2.73) | 3.67 (±2.35) | 3.91 (±3.192) |
| Permanent employees (female) | 2.33 (±1.77) | 1.55 (±3.02) | 0.24 (±0.59) | 1.79 (±1.55) | 2 (±1.90) | 1.58 (±2.02) |
| Occasional employees (male) | 2.76 (±2.21) | 13.2 (±17.38) | 1.71 (±60) | 1.32 (±2.02) | 4.67 (±5.10) | 4.72 (±9.17) |
| Occasional employees (female) | 2.19 (±2.098) | 12.2 (±17.27) | 9.57 (±8.48) | 2.79 (±1.93) | 1.62 (±2.01) | 5.67 (±9.58) |

We identified three categories of land use: land available, land harnessed, and land secured. In the five communes of the department of Borgou, i.e., the study area, the average

amount of land available for agricultural entrepreneurial activities was 17.74 (±15.117) ha. Of this land available, 14.41 (±12.89) ha was harnessed for agricultural entrepreneurial activities, representing 81.22% of the land available. The land available is the land registered by the customary regulations as land rights. According to UN-Habitats (2008), this land is allocated by customary authorities and is espoused for conflict. This proportion is above that of the Borgou department, where the proportion of land harnessed per hectare of land available is 53.9% [48]. Furthermore, 62.90% of the land available is secured (11.16 ha (± 12.296)) through possession of the land paper (ADC or land title). Permanent and seasonal employment is available on the farms of agricultural entrepreneurs. The number of employees differs according to sex. The number of permanent male employees on the farms of agriculture entrepreneur respondents varied from three to five, and the number of women varies from one to two. According to the period and extent of activity on the farm, the number of occasional employees (men) ranged from one to 13, and that of women ranged from two to 12.

### 3.6. The Effects of Accessing Land Mode Factors on Agricultural Entrepreneurial Activities

In this section, we address the effects of accessing land mode factors on agricultural entrepreneurial activities in rural areas, which raise the following hypotheses:

**H0**: *Sociodemographic and land access mode factors have no significant effect on agricultural entrepreneurial activities.*

**H1**: *Sociodemographic and land access mode factors have a significant effect on agricultural entrepreneurial activities.*

The sociodemographic factors identified that can influence agricultural entrepreneurial activities in this study were gender, age, and education level, while the land access mode factors were occupation, heritage, donation, and sharecropping (Table 6).

**Table 6.** The *t*-test of the effect of land access mode factors on agricultural entrepreneurial activities.

| Variables/Factors | *t* | df | Sig. |
|---|---|---|---|
| Gender | 27.523 | 101 | 0.000 * |
| Age | 46.327 | 101 | 0.000 * |
| Education level | 25.673 | 101 | 0.000 * |
| Occupation | 3.126 | 101 | 0.002 * |
| Heritage | 20.349 | 101 | 0.000 * |
| Donation | 3.494 | 101 | 0.001 * |
| Sharecropping | 1.749 | 101 | 0.083 |

* $p \leq 0.05$.

The results show that the *t*-test value for gender was 27.523 while that of age was 46.327, and that of education level was 25.673. However, looking at the Sig. (two-tailed) column, it can be seen that the values of gender, age, and education level were all 0.000, i.e., lower than the standard significance level of 0.05, indicating that the null hypothesis was rejected. As a result, with a 95% confidence level, we can say that gender has a significant effect on agricultural entrepreneurial activities. This significant effect of gender on agricultural entrepreneurial activities is corroborated by the fact that 70% of women are engaged in agriculture (World Bank, 2016), and women are actively involved in food systems in several fundamental functions, i.e., growing and managing crops, livestock, agribusinesses, and food retailing, as well as preparing food for their families [3]. In the department of Borgou, this is also reflected in the descriptive statistics, with a relatively large proportion of our female respondents (11.8% of all female respondents) reporting the intention to be self-employed (Table 3). This result corresponds with modern trends suggesting that entrepreneurship has become more popular among women (see, e.g., [49–51]). Women are credited with playing leading roles in facilitating the introduction of new practices and conceptions on the farm, thus acting as important innovators [52–54]. The average age of

agricultural entrepreneurs was 45.39 (±9.90) years (*t* = 46.327; Sig. (two-tailed) = 0.000). This confirms the result of [55], which revealed that farm growth probability was highest for farmers aged 40–49 years. However, for [56], the taxonomy of entrepreneurial farmers indicated that the "farmer as entrepreneur" was usually younger than 45 years of age. In terms of decline and exit, it has been proven that the younger age group is associated with a lower probability of the business declining and exiting the sector [55,57,58], as younger farmers tend to have more capacity to grow the farm size than older farmers do [59]. The education level had a significant effect on agricultural entrepreneurial activities (*t* = 25.673; Sig. (two-tailed) = 0.000). The education level can cause an ambiguous overall effect; while a higher level of education might benefit the farm's development, well-educated farmers have better job opportunities outside of the farm, which could possibly lead to a reduction in farming activities [60].

As can be seen, the *t*-test value for occupation was 3.126, while that for heritage was 20.349; these values were 3.494 for donation and 1.749 for sharecropping. However, looking at the Sig. (two-tailed) column, it can be seen that the values for occupation, heritage, and donation were respectively 0.002, 0.000, and 0.002, i.e., lower than the standard significance level of 0.05, indicating that the null hypothesis was rejected. As a result, at a 95% confidence level, we can say that respondents' land access mode factors (occupation, heritage, and donation) have a significant effect on agricultural entrepreneurial activities. The *t*-test value for sharecropping was 1.749 with a Sig. (two-tailed) of 0.083. Hence, since the Sig. value was greater 0.05, the null hypothesis was hereby accepted. Accordingly, sharecropping did not have a significant effect on agricultural entrepreneurial activities. Agricultural entrepreneurial activities are considered agriculture investments; thus, the land factors influence agricultural entrepreneurial decisions. According to [61], in Benin, land certification has improved tenure security and stimulated investment in agriculture. Land registration, it is argued, increases credit use through greater incentives for investment in agriculture and reduces incidences of land disputes [62,63]. The resulting legal tenure also influences investments in fixed inputs such as machinery, which are important for enhancing productivity [64].

### 3.7. The Effects of Socioeconomic Factors on Agricultural Entrepreneurial Activities

Agricultural entrepreneurial activities encompass all aspects of food systems (food supply chains, biodiversity, the environment, and consumption). According to [16], rurality and the entrepreneurial process form a dense, complex, and dynamic network of mutual influences. The socioeconomic factors identified in this study included financial support from parents or friends, land availability, experience in the agriculture sector, microfinance or bank loans, autonomy decisions, profitability of activity, and farm government policy (Table 7). The effects of socioeconomic factors on agricultural entrepreneurial activities in rural areas raised the following hypotheses:

**H0**: *Socioeconomic factors have no significant effect on agricultural entrepreneurial activities.*

**H1**: *Socioeconomic factors have a significant effect on agricultural entrepreneurial activities.*

**Table 7.** The *t*-test of the effect of socioeconomic factors on agricultural entrepreneurial activities.

| Variables/Factors | *t* | df | Sig. |
|---|---|---|---|
| Financial support from parents or friends | 4.335 | 101 | 0.000 * |
| Land availability | 6.795 | 101 | 0.000 * |
| Agriculture experience | 5.575 | 101 | 0.000 * |
| Loan from microfinance or bank | 5.878 | 101 | 0.000 * |
| Autonomy decision | 25.196 | 101 | 0.000 * |
| Activity profitability | 27.523 | 101 | 0.000 * |
| Farm government policy | 1.421 | 101 | 0.158 |

* $p \leq 0.05$.

The table shows that the *t*-test result for financial support from parents or friends was 4.335 while that for land availability was 36.885; these values were 5.575 for agricultural

experience, 5.878 for loan from a microfinance or bank, 25.196 for autonomy decision, and 33.420 for activity profitability. However, looking at the Sig. (two-tailed) column, it can be seen that all values were 0.00, i.e., lower than the 0.05 confidence limit. Hence, the null hypothesis was hereby rejected, while the alternate hypothesis was accepted. As a result, with a 95% confidence level, we can say that the respondents' socioeconomic factors (financial support from parents or friends, land availability, experience in the agriculture sector, loan from microfinance or bank, autonomy decision, and profitability of activity) had a significant effect on agricultural entrepreneurial activities. According to the literature, family involvement in the business supports the decision to continue farming [57,65,66] and to expand the business [67]. Manolova et al. (2019) showed that family financial support helps the young entrepreneur overcome capital market voids [63]. The *t*-test value for farm government policy was 1.421 with a Sig. (two-tailed) equal to 0.158. Hence, since the Sig. value was greater than 0.05, the null hypothesis was hereby accepted. Accordingly, the farm government policy had a significant effect on agricultural entrepreneurial activities.

## 4. Conclusions and Recommendations

The land is an essential factor for agricultural entrepreneurial activities and sustainable food production. This study examined the land rights and the land tenure security according to the domain and code of land law in Benin. The different types of land documents ascribed in land law in Benin depict the land rights and the land tenure security following UN-Habitats. The results reveal that the land has customary regulations and is not secure land, reflecting illegal occupation, which results in the most conflicts and corruption. Customary regulation still serves as the land document for 29.5% of the land exploited by agricultural entrepreneurs. This represents the threat of sustainable food production. The contract type is recognized by authorities and grants use rights to agriculture entrepreneurs, which can be disrupted at any moment. The certificate of customary ownership reflects the registered state land for the land rights. This land document provides agricultural entrepreneurship with the legal title and registered record, but not full security for the land exploited. The land title indicates the absence of corruption and conflicts. Only 18% of the land exploited by agricultural entrepreneurs has a land title.

According to the *t*-test value, land access mode factors (occupation, heritage, and donation) have a significant effect on agricultural entrepreneurial activities. This confirms that land tenure security is a crucial factor for agricultural entrepreneurial activities and a key component for sustainable food production and biodiversity protection. Socioeconomic factors (financial assistance from parents or friends, land availability, agricultural experience, microfinance or bank loans, autonomy decision, and activity profitability) also have a significant impact on agricultural entrepreneurial activities.

On the basis of the findings of the study, some recommendations are proposed.

The government should assess the implications and impact of land law because of the legal dualism created by the coexistence of customary and modern rights, the vulnerability of legal land security, and land information management. In practice, the impreciseness of these two types of rights leads to land insecurity, which makes the land vulnerable and subject to all sorts of misappropriation.

Agriculture entrepreneurs should protect their investments by obtaining a land title document for their land.

Other partners, who intervene in food security systems, must facilitate the process of accessing the land security documents because land tenure security safeguards agricultural entrepreneurship and sustainable food production, which is the basis of food security, job creation, and reducing poverty.

All stakeholders in land tenure systems should address, in particular, women's access to agricultural land despite the dominant culture's discrimination, as the majority of women are still in the agriculture sector and at the center of food system production.

**Author Contributions:** Conceptualization, B.D.A. and S.A.A.; methodology, B.D.A.; validation, I.G.A., B.C.P.O. and M.N.B.; formal analysis, B.D.A.; writing—original draft preparation, B.D.A.; writing—review and editing, B.D.A., I.G.A. and S.A.A.; visualization, B.C.P.O.; supervision, M.N.B.; project administration, M.N.B.; funding acquisition, B.C.P.O. All authors have read and agreed to the published version of the manuscript.

**Funding:** This research received no external funding. The funding of the publication costs for this article has kindly been provided by the responsible land policy project of GIZ in Benin (Promotion d'une Politique Foncière Responsable: ProPFR/GIZ).

**Acknowledgments:** The case study is based on the activities of the responsible land policy project of GIZ in Benin, in collaboration with Laboratoire Société–Environnement (LaSEn) (University of Parakou) and Uganda Martyrs University. The responsible land policy project is implemented by Deutsche Gesellschaft für Internationale Zusammenarbeit (GIZ) GmbH (www.giz.de).

**Conflicts of Interest:** The authors declare no conflict of interest.

**Disclaimer:** The content of the publication reflects the personal views of the authors and not those of GIZ.

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
