# Peer review of "Access to Land for Agricultural Entrepreneurial Activities in the Context of Sustainable Food Production in Borgou, according to Land Law in Benin"

_land, doi:10.3390/land11091381_

Round 1

Reviewer 1 Report

Dear authors

Here are some proposals for manuscript improvement.

As this is a scientific paper, ceratin rules on the manuscript structure apply:

1. Introduction

2. Methods and Materials

3. Results and Discussion

4. Conclusions

Please move current chapters 2. and 3. under chapter 2. Methods and Materials as:

2.2 Agriculture entrepreneurial activity

2.3 Conceptual framework

Renumber chapter and sub-chapter that follow.

Line 165: Empirical survey - Please give more details. What kind of questions, and how many were asked. A period when the survey was conducted. I propose that survey is added to the manuscript as supplement material.

Line 193: Add a subchapter, at the end of Methods, on Uncertainties and Shortcomings of the study.

Line 278, Figure 4, text. "land paper"? please use "land document type" or "type of land document."

Line 282-283: add currency exchange rate and date

Figure 4: unify wording with figure 3 and text - land title or land certificate. Add units to the y-axis (%)

Line 343: explain what is "Agri-food transformation" sector. Do you mean crop processing? Crop processing is crucial in converting raw harvested agricultural products into valuable, marketable products. In addition, post-harvest activities can make a big difference to the financial impact of small-scale agriculture.

Table 4: are these numbers meant as average per farm. You have to mark it somewhere in the table.

Line 386: "We can say that gender has a significant effect on agricultural entrepreneurial activities." Do you think you will not be entrepreneurial if you are a woman? I believe that a discussion about deeply rooted traditional roles would be appropriate.

Line 401: "to grow" in what?

Line 404-406:  Discussion about land availability would be more proper. Is there enough land, or are farmers competing regarding land ownership, tenancy and cultivation?

Line 428: "Some environmental factors identified in this study include financial support from parents or friends, land availability, experience in the agriculture sector, microfinance or bank loans, autonomy decisions, the profitability of activity, and farm government policy (table 6)". These are not environmental factors. It would be better to call them socio-economic factors. Change in text and Table 6.

Line 461: Please add a sub-chapter on Recommendations to further support entrepreneurial activity. Based on your results, you must propose how the system and general agricultural land policy could further support farmers by stabilising land ownership.

Line 462-483: Please rephrase the text by including in the text answers to the following questions:

Why is this research unique?

What are the shortcomings/uncertainties of this research?

What did the scientific community learn out of it?

What are the benefits/recommendations for stakeholders (farmers, communes)?

What are the recommendations for policymakers/legislators?

Future work?

Author Response

As this is a scientific paper, certain rules on the manuscript structure apply:

  1. Introduction
  2. Methods and Materials
  3. Results and Discussion
  4. Conclusions

Please move current chapters 2. and 3. under chapter 2. Methods and Materials as:

2.2 Agriculture entrepreneurial activity

2.3 Conceptual framework

Renumber chapter and sub-chapter that follow.

Thank you for your comments. The framing of the paper has been addressed and we have removed the agriculture entrepreneurial activity et the conceptual framework under methods and materials and added the paragraph on the recommendations

Line 165: Empirical survey - Please give more details. What kind of questions, and how many were asked. A period when the survey was conducted. I propose that survey is added to the manuscript as supplement material.

Thank you for the comments and suggestions. The empirical survey was focused on the types of land documents possessed by the agriculture entrepreneurs and their land tenure security following the domain and code of land law in Benin. The survey also examines the outputs of agricultural entrepreneurial activities (activity sector, land harness, job creation, etc.) and the factors influencing this activity. It was carried out in November 2020 with the collaboration of the GIZ institution

Line 193: Add a subchapter, at the end of Methods, on Uncertainties and Shortcomings of the study.

Thank you for the comments. We have introduced a paragraph that frames the complexity of analyzing the agricultural entrepreneurial activity and the importance of one sample T-test.

Line 278, Figure 4, text. "land paper"? please use "land document type" or "type of land document."

This has been corrected.

Line 282-283: add currency exchange rate and date

1 $ = 630.853 FCFA on May 15, 2022

Figure 4: unify wording with figure 3 and text - land title or land certificate. Add units to the y-axis (%)

This has been corrected.

Line 343: explain what is "Agri-food transformation" sector. Do you mean crop processing? Crop processing is crucial in converting raw harvested agricultural products into valuable, marketable products. In addition, post-harvest activities can make a big difference to the financial impact of small-scale agriculture.

Thank you. The Agri-food transformation means precisely crop processing. This has been corrected and unified in all the text.

Table 4: are these numbers meant as average per farm. You have to mark it somewhere on the table.

In table 4 the average numbers mean the average land area per farm and the average number of employees per agricultural entrepreneur. This has been corrected.

Line 386: "We can say that gender has a significant effect on agricultural entrepreneurial activities." Do you think you will not be entrepreneurial if you are a woman? I believe that a discussion about deeply rooted traditional roles would be appropriate.

This significant effect of gender on agricultural entrepreneurial activities is corroborated by the fact that 70% of women are engaged in agriculture (World Bank, 2016), and women are actively involved in food systems in several fundamental functions: growing and managing crops, livestock, agribusinesses, and food retailing, and additionally, in preparing food for their families (Hodson et al., 2021).

Line 401: "to grow" in what?

As younger farmers tend to have more capacity to grow the farm size than older farmers do. This has been corrected.

Line 404-406:  Discussion about land availability would be more proper. Is there enough land, or are farmers competing regarding land ownership, tenancy and cultivation?

The land available is the land registered by the customary regulations as the land right. According to UN-Habitats (2008), this land is allocated by customary authorities and is espoused for conflict.

Line 428: "Some environmental factors identified in this study include financial support from parents or friends, land availability, experience in the agriculture sector, microfinance or bank loans, autonomy decisions, the profitability of activity, and farm government policy (table 6)". These are not environmental factors. It would be better to call them socio-economic factors. Change in text and Table 6.

This has been corrected.

Line 461: Please add a sub-chapter on Recommendations to further support entrepreneurial activity. Based on your results, you must propose how the system and general agricultural land policy could further support farmers by stabilising land ownership.

Thank you for the comments and suggestions. We have tried to tie the findings and discussions to bring clarity to the conclusions and added the paragraph of the recommendations 

Line 462-483: Please rephrase the text by including in the text answers to the following questions:

Why is this research unique?

What are the shortcomings/uncertainties of this research?

What did the scientific community learn out of it?

What are the benefits/recommendations for stakeholders (farmers, communes)?

What are the recommendations for policymakers/legislators?

Future work?

Thank you for the suggestions to improve the frame and the findings. As you can see, we have made efforts to clarify the uniqueness of this study by providing insights to enhance our paper

Reviewer 2 Report

The article is well written, and presents novel data. However, there is room for improvement, as some sections need to be restructured, the introduction is not specific enough in terms of describing the aim of the research, the methods could be more specific in terms of the data collection and analysis, the results and discussion section could be split (and thus, the analysis could be deeper), and the conclusions sections needs improvement.

Please find here my line-by-line review.

47. Please avoid redundancy: "The Republic of Benin is a former colony..."

51. "Several" is vague, I suggest "diverse" or "clashing".

57. Should we know about these laws? Could they be mentioned?

57. New paragraph from "The execution..."

59. (onwards ANDF, by its acronym in French).

60. Remove brackets.

62. the digital cadastre. If the authors would like to keep the original in French, I suggest to use italics for all foreign words.

68. What do the authors refer to as "land system security"? The term is confusing and not explained further.
It could refer to:
1) Tenure security, which goes in line with the rest of the statement.
2) Cultivated Land Systems Security (CLSS), related to the adequate provision of cultivated land.
3) Land systems, as in the processes and activities related to the human use of land.

86. The basic concept of agriculture is not necessary.

87. Remove "as a concept".

95-96. I do not find the basic concept of entrepreneurship necessary.

103. Please insert the author, not just the  reference.

111. The term "land use" could lead to confusion.
For land professionals (e.g. land managers, land economists, agriculturalists, spatial planners), land use is a socio-economic description of land in different categories (agricultural, residential, leisure, commercial, industrial and transportation). The subtle difference between land use and zoning is that land use is the way that people adapt land to suit their needs, while zoning is how the government regulates the land.
Land use changes have a direct impact on development and value (also in human and environmental health).
However, I have the impression that the authors refer to something different here, more in the direction of to what is extent is the available land used. The term is not incorrect, as "land use" could be a generic term. To avoid confusions among land professionals, I suggest to consider "the use of land" instead of "land use".

118. Is this in reference to a specific land conflict? Please describe.
Even if the authors refer to land conflicts in general across Africa, I suggest to be more specific. There are different types of land conflicts that vary from one country to the other and are context-related: land use, land subdivision, land ownership, communal violence, weak governance, land grabbing, among many others.

124-126. New paragraph:
"The land debate in Benin began in the 1990s..."

126. Which and when did the "last" land reform take place? The information in the current form is vague.

128. A more accurate translation would be a "centralised digital cadastre".
Cadastre is a tool for land administration.

186. To this point, there is no clarity on the type of data gathered in the interviews. I thus suggest to expand the methodology section.

236. The diagram showing the different levels of tenure security could be enriched by UN Habitat's continuum of land rights, in which we can see a wider spectrum of informal to formal land rights.

266. The diagram features articles 351, 354, 352 and 376 in correlation with perceived tenure security. The description of the diagram should go deeper into the different certificates, the laws/policies that support it, and the perceived security, level by level.

282. Is the range between 30-50,000 or 30,000-50,000?

286. Insert name of author, not just the reference.

297-300. Figure 4 could be improved by removing the lines that link the land papers percentages across communes. I suggest to simply leave the coloured bars and figures.

322. The gender imbalance in agripreneurship could be an evidence of inequalities in the access to land, capacity building, decision-making, and funds/loans for resources and investments. It could be a consequence of non-inclusive land policies (i.e. written in masculine terms instead of neutral, a low number of women in decision-making positions, discrimination) and/or patriarchal customary practices, which hinder the exercise of women's land rights.
It could also be related to the sampling method.
I suggest to take a closer look to Kalalé in order to find the main difference that favour female agripreneurship.

341-345. As mentioned before, it would be interesting to explore why and how the gender gap in agripreneurship has not only closed but inverted in Kalalé. This could be an indicator of something as positive as inclusive policies, or as negative as armed conflict and forced displacement.
Up to this point, the results are not being discussed nor analysed.

367. This is the point when the discussion starts. I thus suggest to separate the results and discussion sections.

426. Include author, not just the reference.

462. The conclusion section needs further development.
It limits to summarise the discussion, instead of emphasising the main findings, the adequateness of the methodology, the corroboration of the hypothesis (or answer to the research question), the recommendations based on the findings, and the recommendations for further research.

464-466. Redundant.

513. Typo

The line-by-line review could be easier to follow with the highlighted and strikethrough text in the attached file.

Author Response

  1. Please avoid redundancy: "The Republic of Benin is a former colony..."
  2. "Several" is vague, I suggest "diverse" or "clashing".

Thank you for the comments. This has been corrected.

  1. Should we know about these laws? Could they be mentioned?

Thank you for your comments. We have reworked the introduction and improved the focus on land law in Benin. The new land law encompasses, art. 347 to art. 378, the customary and rural land rights and the justice and administrative functions of rural land management (PRB, 2017). We have added the table of chronological evolution of land policies in the Benin Republic after 1990

  1. New paragraph from "The execution..."
  2. (onwards ANDF, by its acronym in French).
  3. Remove brackets.
  4. the digital cadastre. If the authors would like to keep the original in French, I suggest to use italics for all foreign words.

Thank you for the suggestion. We have identified and corrected this. We hope this is now clearer.

  1. What do the authors refer to as "land system security"? The term is confusing and not explained further.

It could refer to:

1) Tenure security, which goes in line with the rest of the statement.

2) Cultivated Land Systems Security (CLSS), related to the adequate provision of cultivated land.

3) Land systems, as in the processes and activities related to the human use of land.

Thank you for the suggestions on improving the introduction. As you can see, we have made efforts to clarify the introduction and the concept of "land tenure security" is appropriate in our case

  1. The basic concept of agriculture is not necessary.

95-96. I do not find the basic concept of entrepreneurship necessary.

Thank you for the suggestion. We have clarified the concepts for the uniqueness and relevance of the study context.

  1. Please insert the author, not just the reference.
  2. The term "land use" could lead to confusion.

For land professionals (e.g. land managers, land economists, agriculturalists, spatial planners), land use is a socio-economic description of land in different categories (agricultural, residential, leisure, commercial, industrial and transportation). The subtle difference between land use and zoning is that land use is the way that people adapt land to suit their needs, while zoning is how the government regulates the land.

Land use changes have a direct impact on development and value (also in human and environmental health).

However, I have the impression that the authors refer to something different here, more in the direction of to what is extent is the available land used. The term is not incorrect, as "land use" could be a generic term. To avoid confusions among land professionals, I suggest to consider "the use of land" instead of "land use".

Thank you for the comments. This has been corrected.

  1. Is this in reference to a specific land conflict? Please describe.

Even if the authors refer to land conflicts in general across Africa, I suggest to be more specific. There are different types of land conflicts that vary from one country to the other and are context-related: land use, land subdivision, land ownership, communal violence, weak governance, land grabbing, among many others.

The land conflict appears in Benin for a long-term loan or if one of the actors dies. Borrowers can in bad faith claim to have received the land as a gift or to have acquired it by other means. There is also a land conflict for long-term pledges where key players may have died without leaving a written record attesting to the pledge. We have clarified this by using the UN-Habitats book.

124-126. New paragraph:

"The land debate in Benin began in the 1990s..."

  1. Which and when did the "last" land reform take place? The information in the current form is vague.
  2. A more accurate translation would be a "centralised digital cadastre".

Cadastre is a tool for land administration.

Thank you for the suggestion. We have now addressed this and improved the section

  1. To this point, there is no clarity on the type of data gathered in the interviews. I thus suggest to expand the methodology section.

Thank you for the recommendations. We have added the studies and enhanced the contributions.

  1. The diagram showing the different levels of tenure security could be enriched by UN Habitat's continuum of land rights, in which we can see a wider spectrum of informal to formal land rights.
  2. The diagram features articles 351, 354, 352 and 376 in correlation with perceived tenure security. The description of the diagram should go deeper into the different certificates, the laws/policies that support it, and the perceived security, level by level.

The diagram have been improved following the land rigths and land tenure security of UN-Habitats.

According to UN-Habitat (2018), tenure security still tends to be strictly defined in more statutory forms of legal documentation, such as individual land titles. However, the customary regulations recognized by the land and domain code law (art. 354) don't guarantee the security of land for agricultural entrepreneurial activities. Since colonial times, customary land tenure was not thought to provide adequate tenure security, thereby discouraging investment and negatively affecting agricultural productivity Swynnerton, 1954; Wilson, 1971). The customary land tenure corresponds to the illegal occupation where there are the most conflicts and corruption. The contract type (Art. 354) is recognized by authorities (legitimacy axe) and grants use rights to agriculture entrepreneurs. The certificate of customary ownership (art. 352) reflects the registered state land for the land rights. This land document provides agriculture entrepreneurship the legal title and registered a record for the land tenure security. The land title (art. 376) indicates the absence of corruption and conflicts. It is the high level of land tenure security that corresponds to agricultural entrepreneurship and security investment for sustainable food production

  1. Is the range between 30-50,000 or 30,000-50,000?
  2. Insert name of author, not just the reference.

297-300. Figure 4 could be improved by removing the lines that link the land papers percentages across communes. I suggest to simply leave the coloured bars and figures.

Thank you for your suggestion. This has been reviewed

  1. The gender imbalance in agripreneurship could be an evidence of inequalities in the access to land, capacity building, decision-making, and funds/loans for resources and investments. It could be a consequence of non-inclusive land policies (i.e. written in masculine terms instead of neutral, a low number of women in decision-making positions, discrimination) and/or patriarchal customary practices, which hinder the exercise of women's land rights.

It could also be related to the sampling method.

I suggest to take a closer look to Kalalé in order to find the main difference that favour female agripreneurship.

Thank you for your suggestion. This has been reviewed

341-345. As mentioned before, it would be interesting to explore why and how the gender gap in agripreneurship has not only closed but inverted in Kalalé. This could be an indicator of something as positive as inclusive policies, or as negative as armed conflict and forced displacement.

Up to this point, the results are not being discussed nor analysed.

  1. This is the point when the discussion starts. I thus suggest to separate the results and discussion sections.
  2. Include author, not just the reference.
  3. The conclusion section needs further development.

It limits to summarise the discussion, instead of emphasising the main findings, the adequateness of the methodology, the corroboration of the hypothesis (or answer to the research question), the recommendations based on the findings, and the recommendations for further research.

Thank you for the comments and suggestions. We have tried to tie the findings and discussions to bring clarity to the conclusions and added the paragraph of the recommendations 

464-466. Redundant.

This has been reviewed and removed from the paragraph.

  1. Typo

The line-by-line review could be easier to follow with the highlighted and strikethrough text in the attached file.

Thank you for your suggestion. This has been reviewed

Reviewer 3 Report

Title:  …in Borgou (Benin). The study is not all about Benin, only for Borgou Department.

Section 4.2. Some figures are less to define the representative level of the number of samples done: How many villages are in de department of Borgou? How many total agricultural entrepreneurs are in the 5 communes?

Tabel 4. Left Column: Units for male, female: numbers? Please check that all male/female words start with capital letters.

Table 5. Usually, 2 decimal figures are enough. It does not necessarily show the complete table generated by SPSS statistical package.  Variables, t figures, number of samples (better than df), and Significant values are sufficient.  All the variables, except “Sharecropping” are statistically significative, then the discussion about this fact should be holistic: in the present way the text seems reflects that Gender, Age and Educational level are more significant than Occupation, Heritage and Donation. A deep discussion should be done.

The figures in the Tables should not be repeated in the text: only reference the significance or not, (p=0.05) will be right.

The same for Table 6.

Line 391: Please, check the reference citation in the text 

Section 5.7. The use of “environmental factors” to refer to the several variables for entrepreneurial activities maybe produce a misunderstanding.

How do the variables quantification (financial support, land availability, agricultural experience…)  were done to compare them by a one-sample test?

Results, and Discussion, should be analyzed most deeply.  A Factorial Analysis Test maybe adds more information about the different weights for each variable and shows variables associations to help the Discussion and Conclusions processes.

Author Response

Title:  …in Borgou (Benin). The study is not all about Benin, only for Borgou Department.

Thank you for your suggestion. We have identified and corrected this. We hope this is now clearer.

Section 4.2. Some figures are less to define the representative level of the number of samples done: How many villages are in de department of Borgou? How many total agricultural entrepreneurs are in the 5 communes?

Thank you for the comments. The study was carried in the 5 communes' beneficiaries of the Responsible Land Policy Project of GIZ (Promotion d’une Politique Foncière Responsible: ProPFR/GIZ). We have tried to find the number of village and agricultural entrepreneurs in the official document of the country which is not available. 

Tabel 4. Left Column: Units for male, female: numbers? Please check that all male/female words start with capital letters.

 Thank you for the comments. This has been corrected.

Table 5. Usually, 2 decimal figures are enough. It does not necessarily show the complete table generated by SPSS statistical package.  Variables, t figures, number of samples (better than df), and Significant values are sufficient.  All the variables, except “Sharecropping” are statistically significative, then the discussion about this fact should be holistic: in the present way the text seems reflects that Gender, Age and Educational level are more significant than Occupation, Heritage and Donation. A deep discussion should be done.

The figures in the Tables should not be repeated in the text: only reference the significance or not, (p=0.05) will be right.

The same for Table 6.

Line 391: Please, check the reference citation in the text

Section 5.7. The use of “environmental factors” to refer to the several variables for entrepreneurial activities maybe produce a misunderstanding.

How do the variables quantification (financial support, land availability, agricultural experience…)  were done to compare them by a one-sample test?

Thank you for the comments. This has been corrected.

Results, and Discussion, should be analyzed most deeply.  A Factorial Analysis Test maybe adds more information about the different weights for each variable and shows variables associations to help the Discussion and Conclusions processes

Thank you for the comments and suggestions. We have tried to tie the findings and discussions to bring clarity to the conclusions and added the paragraph of the recommendations 

Round 2

Reviewer 2 Report

The authors have significantly improved the manuscript, and I believe it is ready for publication.

I suggest the editor to look at the following minor mistakes:

39-41. Verify reference 4.

70. Insert author's name, not just reference.

73-74. Insert author's name, not just reference.

108-109. Reference for the Land and Domain Code Law?

150. Missing punctuation mark.

180. The page number is not necessary.

227. Invert languages: English in text and French in brackets.

Figure 3. Excellent graphic!

349. Insert author's name, not just reference.

428. UN Habitat

470. Insert author's name, not just reference.

471. Insert author's name, not just reference.

475-476. This is a complement, not a complete sentence.

480. Remove period between "farm" and "and".

493. Insert author's name, not just reference.

500. Insert author's name, not just reference.

Reviewer 3 Report

Changes were done in a  proper way